# Serpentine Enhances Insulin Regulation of Blood Glucose through Insulin Receptor Signaling Pathway

**DOI:** 10.3390/ph16010016

**Published:** 2022-12-22

**Authors:** Yinghao Wang, Guanfu Liu, Xutao Liu, Minhua Chen, Yuping Zeng, Yuyan Li, Xiaoyun Wu, Xuanjun Wang, Jun Sheng

**Affiliations:** 1Key Laboratory of Puer Tea Science, Ministry of Education, Yunnan Agricultural University, Kunming 650201, China; 2Scientific Observing and Experimental Station of Tea Resources and Processing in Yunnan, Ministry of Agriculture, Kunming 650201, China; 3Department of Science, Yunnan Agricultural University, Kunming 650201, China; 4Yunnan Research Institute for Local Plateau Agriculture and Industry, Kunming 650201, China

**Keywords:** serpentine, insulin receptor, AMPK, GLUT4, T2DM

## Abstract

Insulin sensitizers targeting insulin receptors (IR) are a potential drug for the treatment of diabetes. Serpentine is an alkaloid component in the root of *Catharanthus roseus* (L.) G. Don. Serpentine screened by surface plasmon resonance (SPR) technology has the ability to target IR. The objective of this study was to investigate whether serpentine could modulate the role of insulin in regulating blood glucose through insulin receptors in cells and in animal models of diabetes. SPR technology was used to detect the affinity of different concentrations of serpentine with insulin receptors. The Western blotting method was used to detect the expression levels of key proteins of the insulin signaling pathway in C2C12 cells and 3T3-L1 cells as well as in muscle and subcutaneous adipose tissue of diabetic mice after serpentine and insulin treatment. Diabetic mice were divided into four groups and simultaneously injected with insulin or serpentine, and the blood glucose concentration and serum levels of insulin, glucagon, and C-peptide were measured 150 min later. mRNA levels of genes related to lipid metabolism and glucose metabolism in liver, muscle, and subcutaneous adipose tissue were detected by RT-PCR. Serpentine was able to bind to the extracellular domain of IR with an affinity of 2.883 × 10^−6^ M. Serpentine combined with insulin significantly enhanced the ability of insulin to activate the insulin signaling pathway and significantly enhanced the glucose uptake capacity of C2C12 cells. Serpentine enhanced the ability of low-dose insulin (1 nM) and normal-dose insulin (100 nM) to activate the insulin signaling pathway. Serpentine also independently activated AMPK phosphorylation, thus stimulating glucose uptake by C2C12 cells. In high-fat-diet/streptozotocin (HFD/STZ)-induced diabetic mice, serpentine significantly prolonged the hypoglycemic time of insulin, significantly reduced the use of exogenous insulin, and inhibited endogenous insulin secretion. In addition, serpentine alone significantly increased the expression of *GSK-3β* mRNA in muscle tissue, thus enhancing glucose uptake, and at the same time, serpentine significantly increased glucagon secretion and liver gluconeogenesis. Serpentine enhances the ability of insulin to regulate blood glucose through the insulin receptor, and can also regulate blood glucose alone, but it has a negative regulation mechanism and cannot produce a hypoglycemic effect. Therefore, serpentine may be useful as an insulin sensitizer to assist insulin to lower blood glucose.

## 1. Introduction

Type 2 diabetes is a serious and common chronic metabolic disease with several macrovascular complications, such as cardiovascular disease (CVD), and microvascular complications affecting the kidneys, retina, and nerves [1]. An estimated 536.6 million people have been reported with diabetes worldwide in 2021 and the number is expected to further increase to 783.2 million by 2045 [2]. Regulation of blood glucose and protection of beta-cell function in the early stages of disease may have important implications for preventing subsequent complications of T2DM.

Insulin receptor, a member of the transmembrane ligand-activated receptor and tyrosine kinase family, is composed of two α and two β subunits [3]. Insulin binds to and changes the conformation of the insulin receptor, consequently stimulating the activity of receptor kinases through autophosphorylation of tyrosine residues in the β subunit [4]. The insulin signal is transmitted to downstream kinases, including protein kinase B (AKT) and glycogen synthase kinase (GSK) 3β. Insulin signaling also leads to translocation of glucose transporter 4 (GLUT4) and enhanced glucose uptake [5,6]. Insulin treatment is the first choice for a large proportion of diabetic patients [7] but is associated with several side-effects [8]. The key to successful treatment of diabetes is to develop insulin substitutes and auxiliary plants for reducing adverse effects. Previous studies have identified active ingredients in traditional Chinese herbal medicine that enhance insulin sensitivity through promoting IR autophosphorylation [9,10,11,12]. The use of natural compounds as insulin sensitizers may thus have significant benefits in the treatment of diabetes.

*Catharanthus roseus* (L.) G. Don is a traditional antidiabetic drug widely used worldwide and its therapeutic effects are attributed to constituent alkaloids [12]. In many regions, the leaves or whole plant are decocted for treatment of diabetes [13] and glycemic activity of the ethanolic extract has additionally been reported [14,15,16]. Two vinca alkaloids, vincristine and vinblastine isolated from *Catharanthus roseus* (L.) G. Don, have been extensively studied for their anti-cancer and anti-diabetic potential [17]. Serpentine is an alkaloid component in the root of *Catharanthus roseus* (L.) G. Don [18] used in the treatment of hypertension and heart disease [19,20]. In a previous study using surface plasmon resonance (SPR) to screen more than 4000 small molecule natural products, our group showed that serpentine could bind the extracellular domain of the insulin receptor with a high response value.

In this study, we focused on the potential anti-diabetic effect of serpentine and underlying mechanisms. SPR technology was employed to determine the affinity of serpentine for IR and its effects on binding of insulin to IR. In vitro, serpentine enhanced insulin-mediated tyrosine autophosphorylation of the IRβ subunit and significantly enhanced the ectopic GLUT4 level in C2C12 myoblasts and 3T3-L1 primary adipocytes. Treatment with serpentine alone stimulated GLUT4 translocation and enhanced glucose uptake. In vivo, serpentine acted as an insulin sensitizer, enhancing the effect of insulin (0.5 U/kg) in an HFD/STZ-induced type 2 diabetes mouse model. Taken together, our findings indicate that serpentine contributes to the antidiabetic activity of insulin mainly in an insulin-receptor-dependent manner.

## 2. Results

### 2.1. Cytotoxicity of Serpentine and Affinity of Serpentine for Insulin Receptor

Interactions of insulin or other drugs with insulin receptors can be detected using surface plasmon resonance (SPR) [21]. TLK16998 is reported to act as an insulin sensitizer and to activate insulin receptor subunit tyrosine kinase via binding insulin receptor in the presence of insulin [22]. Based on data from earlier studies, we determined the binding affinity of serpentine to insulin receptor using the SPR assay. Serpentine bound the insulin receptor with an affinity of 2.883 × 10^−6^ M (Figure 1B). To further establish whether serpentine could be developed as a potential clinical drug, a cytotoxicity test was performed. Our results showed that serpentine did not affect the viability of 3T3-L1 and C2C12 cells at concentrations lower than 32 μM (Figure 1C,D). Accordingly, we propose that serpentine exerts its effects via binding to the insulin receptor and does not exert cytotoxic effects at low doses.

### 2.2. Effects of Serpentine on the Insulin Signaling Pathway Activated by Low-Dose Insulin

Next, we explored whether the interactions of serpentine with IR affect the classical insulin signaling pathway. Tyrosine phosphorylation of the IR-β subunit and subsequent increase in phosphorylation of endogenous substrate AKT are indicators of the enhanced effect of insulin [23]. Differentiated and mature 3T3-L1 and C2C12 cells were treated with 100 nM insulin and 10 µM serpentine (Figure 2A,B), respectively. Insulin (100 nM) induced significant tyrosine phosphorylation of the IR-β subunit as well as phosphorylation of downstream AKT while serpentine alone could not activate the insulin signaling pathway. Notably, however, phosphorylation-based activation of IR-β and AKT was more significant in the presence of serpentine combined with insulin than insulin alone (100 nM) (Figure 2A,B). In cases of severe insulin resistance, nonsense or missense mutations in the extracellular binding domain or intracellular tyrosine binding domain of the receptor have been shown to severely reduce insulin binding. Patients with this type of diabetes require 100 times or more insulin than the typical requirement [24]. 3T3-L1 and C2C12 cells were additionally treated with low-dose insulin (1 nM) and serpentine (10 µM). In 3T3-L1 cells, both low-dose insulin (1 nM) and serpentine (10 µM) led to enhanced insulin signaling, observed as an increase in tyrosine phosphorylation of the IR-β subunit and subsequent phosphorylation of the endogenous substrate AKT (Figure 2C). These effects were more pronounced upon co-treatment with insulin and serpentine. Simultaneous treatment with low-dose insulin (1 nM) and serpentine (10 μM) did not induce tyrosine phosphorylation of the IR-β subunit in C2C12 cells but significantly activated insulin signaling (Figure 2D). These results support the utility of serpentine as an insulin sensitizer to enhance the effects of insulin in vitro.

### 2.3. Effects of Serpentine on Blood Glucose and Hormone Levels in Mice with STZ/HFD-Induced Type 2 Diabetes Treated with Insulin

To further ascertain whether serpentine can enhance the hypoglycemic effect of insulin in vivo, the insulin tolerance test (ITT) was performed in mice with HFD/STZ-induced diabetes. To this end, an HFD/STZ-induced diabetes mouse model was initially established (Figure 3A–C), At the time of the ITT, serpentine (1.5 mg/kg) was simultaneously administered. Combination therapy with serpentine and insulin led to a significantly enhanced hypoglycemic effect and prolonged hypoglycemic time (Figure 3D,E). Mouse serum tests after 150 min of drug treatment revealed that co-treatment with serpentine and insulin induced a significant increase in serum insulin concentration (Figure 3F) and decrease in C-peptide secretion (Figure 3H) while serpentine alone markedly enhanced glucagon secretion (Figure 3G). These results support the activity of serpentine as a sensitizer that effectively enhances the hypoglycemic effect of insulin in vivo.

### 2.4. Effects of Serpentine and Insulin Treatment on the Insulin Signaling Pathway in Target Organs

We further explored whether the serpentine-induced enhancement of insulin sensitivity in vivo is related to activation of the insulin signaling pathway in target organs. Accordingly, expression of key proteins of the insulin signaling pathway in muscle tissue and subcutaneous adipose tissue was examined at 150 min after drug treatment. Notably, phosphorylation of the IR-β subunit and AKT remained significantly enhanced in muscle and subcutaneous adipose tissues in the group co-treated with insulin and serpentine (Figure 4), consistent with the results of blood glucose tests. Our findings indicate that serpentine enhances the hypoglycemic effect of insulin by promoting its activation of the insulin signaling pathway in target organs in vivo.

### 2.5. Effects of Serpentine and Insulin Treatment on Genes Related to Gluconeogenesis, Glycogen Synthesis, and Fat Synthesis

Insulin plays a key role in metabolism, inducing significant inhibition of gluconeogenesis-related genes in the liver [25], along with enhanced glycogen synthesis in muscle tissue [26], and triglyceride uptake and fatty acid synthesis in adipose tissue [27]. Here, we measured mRNA levels in liver, muscle, and subcutaneous adipose tissues of drug-treated mice. In muscle tissue, although expression of *Ugp2* and *Gys* did not differ among groups, *Gsk3β* levels in the serpentine and Ser+INS treatment groups were significantly increased (Figure 5A–C). In subcutaneous fat, compared with the insulin treatment group, the Ser+INS group showed a tendency of decreased expression of the fat-synthesis-related genes *Fas* and *Acc1* (Figure 5D,E). In liver tissue, expression of *G6Pase* in the insulin and Ser+INS groups was significantly inhibited while that of *G6Pase* and *Pepck* in the serpentine treatment group was significantly enhanced (Figure 5F,G). Moreover, expression of *Gys* in the serpentine and Ser+INS groups was markedly inhibited (Figure 5H). These results indicate that serpentine upregulates glucose uptake genes in muscle tissue as well as gluconeogenic genes in liver tissue, which may underlie its lack of hypoglycemic activity in vivo.

### 2.6. Serpentine Promotes Glucose Uptake through the AMPK Signaling Pathway

Insulin-induced GLUT4 translocation is activated by the IR-AKT-AS160 pathway [28]. In our experiments, serpentine treatment alone did not activate phosphorylation of insulin receptor tyrosine kinase but promoted GLUT4 expression (Figure 6A,D). The data indicate that serpentine can act on multiple functional targets to activate glucose uptake in cells. AMPK, a receptor of energy metabolism, improves glucose uptake and insulin resistance and plays a key role in maintaining the energy metabolism balance [29]. In this study, the phosphorylated AMPK protein level was detected in C2C12 cells. Expression of phosphorylated AMPK protein was significantly increased in the insulin, SER, and SER+INS groups compared with the control group (Figure 6A,C). GLUT4 translocation from the cytosol to plasma membrane, critical for glucose metabolism, is promoted by AMPK-AS160 signaling activation [30]. In C2C12 cells, the protein expression patterns of phosphorylated AS160 were consistent with those of GLUT4 and phosphorylated AMPK (Figure 6A,B,D). Meanwhile, the glucose uptake of C2C12 cells treated with serpentine and insulin was consistent with the expression of GLUT4, pAS160, and pAMPK (Figure 6E). Based on the results, we infer that serpentine can also promote the membrane transfer of GLUT4 and glucose uptake by cells through activation of phosphorylated AMPK.

## 3. Discussion

Insulin activates downstream signaling pathways via binding to specific receptors [4] and all insulin receptor domains have the potential to serve as drug targets [31]. Several studies have shown that extracts of *Catharanthus roseus* have hypoglycemic activity and enhance GLUT4 gene expression [32,33]. *Catharanthus roseus* (L.) G. Don contains a variety of alkaloid components with hypoglycemic properties, such as vindogentianine identified by Tiong et al. [34] and other subsequently isolated compounds [35]. With the aid of SPR technology to screen natural small-molecule compounds with affinity for insulin receptor, we identified serpentine from *Catharanthus roseus* (L.) G. Don showing high affinity for the extracellular domain, determined as 2.883 × 10^−6^ M with Biacore S200. Based on this finding, we explored whether serpentine could serve as an IR activator with insulin-sensitizing activity through interactions with the insulin receptor, similar to TLK 16998 [22]. The MTT assay [36] revealed no cytotoxicity of serpentine, even at the highest dose up to 32 µM. Therefore, serpentine can exert its effects through the insulin receptor at concentrations lower than those producing cytotoxicity.

In this study, using insulin as a positive control, we showed that a combination of serpentine and insulin in vitro significantly activated the insulin signaling pathway, indicating that serpentine serves as an enhancer of insulin sensitivity. In C2C12 and 3T3-L1 cells treated with normal doses of insulin (100 nM), insulin-mediated activation of the signaling pathway was significantly increased in the presence of serpentine compared with the insulin group only. However, serpentine alone did not have the ability to activate insulin signaling. In severe insulin resistance, insulin binding is severely reduced due to insulin receptor mutations that cause nonsense or missense substitutions in the extracellular binding domain or intracellular tyrosine-binding domain, and therefore, patients with this type of diabetes require 100 times or more insulin than the typical requirement [24]. In subsequent experiments, we used a low dose of insulin (1 nM) to simulate reduced insulin binding to the receptor and examined whether the presence of serpentine could enhance activation of the signaling pathway mediated by low-dose insulin. Our data showed that serpentine was an effective insulin analog in 3T3-L1 cells at a dose of 10 µM, achieving the same intensity of signaling as low-dose insulin (1 nM). In C2C12 cells, at the same dose of 10 µM, serpentine could not act as an insulin analog to activate signaling but functioned as an insulin sensitizer, indicative of effects on the insulin signaling pathway through other targets. The effect of low-dose insulin (1 nM) on insulin signaling pathway activation was significantly enhanced in the presence of serpentine in C2C12 cells. The expression patterns in different cells may be due to unique cell-type-specific metabolic effects in each target organ. Taking insulin signaling as an example, only IRS1 is involved in muscle tissue while both IRS1 and IRS2 play a role in adipose tissue [37]. Earlier studies have reported that inhibition of cellular protein tyrosine phosphatase activity promotes autophosphorylation of proteins [38]. Our results further support the notion that serpentine acts directly on IR to facilitate and maintain autophosphorylation. Serpentine could not directly activate IR in C2C12 cells, but significantly enhanced phosphorylation and glucose transport ability of IR when combined with low-dose insulin (1 nM). In conjunction with molecular interaction data, these findings suggest that serpentine interacts with IRβ and specifically enhances the effect of low doses of insulin (1 nM) on IRβ self-phosphorylation and subsequent downstream signaling.

Serpentine effectively enhanced the regulation of blood glucose by insulin in our HFD/STZ-induced type 2 diabetes mouse model. Based on an earlier protocol, HFD and low-dose STZ were used to establish the stable type 2 diabetes model [39]. In the insulin and serpentine tolerance test, serpentine alone did not exert a hypoglycemic effect, but significantly enhanced the hypoglycemic effect of insulin in co-treated mice. The combination of serpentine and insulin reduced the C-peptide content and increased the insulin content in sera of mice at 150 min, indicating lower consumption of exogenous insulin. This could be explained by inhibition of endogenous insulin secretion by exogenous insulin along with C-peptide as a concomitant secretion product of endogenous insulin [40,41]. The serpentine-enhanced hypoglycemic effect of insulin in vivo could be related to the increase in the number of insulin receptors activated by serpentine, as discussed earlier. Glucagon, a counterregulatory hormone of insulin, promotes the hepatic glucose output and elevates blood glucose by increasing glycogenolysis and gluconeogenesis and synergistically reducing glucose production and glycolysis through multiple mechanisms [42]. The results of glucagon detection in serum showed that serpentine markedly stimulated glucagon secretion in vivo. However, this effect of serpentine was significantly inhibited upon combination treatment with insulin, consistent with the finding that insulin inhibits glucagon secretion [43]. Our collective results support the utility of serpentine as a promising sensitizer that can enhance the hypoglycemic effect of insulin in vivo.

Blood glucose is regulated by a variety of genes in vivo [44,45]. Detection of genes related to blood glucose regulation in liver, muscle, and subcutaneous adipose tissue of type II diabetic mice treated with insulin and serpentine showed that the combination of serpentine and insulin induced significant *Gsk3β* expression in muscle. Serpentine alone also significantly enhanced *Gsk3β* expression in muscle. Expression of *Gsk3β* promotes glycogen synthesis in tissues and reduces blood glucose [46]. Although serpentine treatment led to higher *Gsk3β* levels in muscle, the alkaloid did not exert a hypoglycemic effect, which could be attributed to significant enhancement of glucagon secretion. Moreover, enhanced expression of *Pepck* and *G6Pase* in the liver after treatment with serpentine alone did not lead to reduced blood glucose [47]. Further studies are warranted to determine why serpentine does not have the ability to exert a hypoglycemic effect.

In this study, serpentine promoted GLUT4 membrane transfer and cellular glucose uptake through phosphorylation of AS160, similar to insulin. Previous studies have shown that glucose uptake by skeletal muscle can be enhanced through multiple pathways, such as translocation of GLUT4 through Ca^2+^ signaling [48,49] and localization of GLUT4 to the plasma membrane through the PI3K-AKT pathway [50]. Activation of pAS160 contributes to membrane translocation of GLUT4 and glucose transport in muscle [28]. Our experiments revealed that serpentine promoted GLUT4 translocation, which was further enhanced upon co-treatment with insulin. Additionally, glucose uptake was increased in the presence of serpentine. Serpentine alone failed to activate the insulin signaling pathway in C2C12 cells compared to insulin alone (100 nM), indicating other targets of serpentine exist that activate glucose uptake in cells, consistent with data from earlier studies on natural compounds supporting multiple targets and pathways [51]. AMPK activation in skeletal muscle enhances glucose uptake [52,53] and many natural compounds promote glucose uptake in an AMPK-dependent manner. For example, bioactive components in *Andrographitis paniculata* have been shown to ameliorate muscle insulin resistance through activating AMPK [54]. Examination of the AMPK signaling pathway after serpentine treatment showed that serpentine not only acts as an insulin sensitizer but also promotes GLUT4 transfer to the plasma membrane and glucose uptake via stimulating AMPK phosphorylation. In addition, serpentine induced high expression of *Gsk3β* in muscle in vivo. However, the other potential pathways by which serpentine regulates cellular glucose uptake remain to be elucidated.

## 4. Materials and Methods

### 4.1. Cell Culture and Treatments

Mouse primary adipocytes, 3T3-L1, and mouse myoblasts, C2C12, were purchased from the Kunming Cell Bank of the Chinese Academy of Sciences (Kunming, China). Cells were maintained in DMEM (Meilunbio, Dalian, China) supplemented with 10% FBS (Biological Industries, Beit Haemek, Israel) and 100 U/mL penicillin and streptomycin (Meilunbio) in a CO_2_ incubator (5% CO_2_) at 37 °C until confluence.

Prior to the experiment, 3T3-L1 cells were cultured and differentiated into adipocytes. After full confluence, cells were incubated with DMEM containing 10% FBS and MDI (IBMX 0.5 mM, DEX 1 μM, and insulin (INS, 5 μg/mL)) for 72 h. The medium was replaced with 10% FBS and insulin (INS, 5 μg/mL) in DMEM for 72 h, followed by DMEM containing 10% FBS every 2 days until differentiation was complete. C2C12 cells were cultured and differentiated into muscle cells. Fully confluent cells were further cultured with DMEM containing 2% HS (Gibco, Carlsbad, CA, USA) for 7 days, with changes every 2 days until differentiation was complete. Prior to drug treatment, cells were starved in serum-free medium for 4 h and treated with insulin (1 nM or 100 nM) (Novo Nordisk, Copenhagen, Switzerland) and serpentine (10 μM) alone, or in combination, for 20 min for further experiments. Serpentine was purchased from Yunnan XiLi Biotechnology Co., LTD (BioBioPha, Kunming, China).

### 4.2. Animals

All animal experiments were approved by the Animal Ethics Committee of Yunnan Agricultural University (No. 202010057). Male C57BL/6J mice were purchased from Cavens Laboratory (Changzhou, China). After one week of acclimation (day 0), all mice were fed a high-fat diet (HFD, 60% fat; D12492, Research Diet) for 16 days. When weights were up to 30 g, mice were injected intraperitoneally with 40 mg/kg streptozotocin (Sigma-Aldrich Co., St Louis, MO, USA) after being starved for 12 h before injection, and feed was provided 30 min after injection. The same method was used for two consecutive days. Two weeks after the first injection (30 days), 30 mg/kg streptozotocin was further injected for establishing a type 2 diabetes model [39]. Animals with random blood glucose levels >17 mmol/L were defined as diabetic-model mice [55]. Diabetic mice (*n* = 24) were divided into four experimental groups: (1) control, receiving an intraperitoneal injection of normal saline, (2) insulin, whereby a dose of 0.5 U/kg of insulin was subcutaneously injected into the abdomen, (3) serpentine, with intraperitoneal injection of serpentine at a dose of 1.5 mg/kg, and (4) Ser+INS, insulin, and serpentine administered at the same doses as the above groups. After 150 min, blood samples were collected from the orbital venous plexus under ether general anesthesia and serum separated for analysis of biochemical parameters. Serum insulin (#90080), C-peptide (#90050), and glucagon (#81518) levels were determined using a kit from Crystal Chem (Crystal Chem Inc., Princeton, NJ, USA). Following blood collection, mice were euthanized and dissected, and liver, muscle, and subcutaneous adipose tissues collected for protein immunization experiments.

### 4.3. Cell Viability Assay

3T3-L1 and C2C12 cells were cultured in 96-well dishes at a density of 2 × 10^5^ cells per mL at 37 °C and 5% (*v*/*v*) CO_2_ for 12 h. Following adherence, cells were incubated with or without serpentine for 24 h and subsequently with methylthiazolyldiphenyl-tetrazolium bromide (MTT; Solarbio, Beijing, China) at a final concentration of 0.5 mg/mL for 4 h. The medium was removed, 150 μL dimethyl sulfoxide (DMSO) was added to each well, and after shaking for 20 min, absorbance at 492 nm was measured using a Flex station 3 multifunctional microplate reader (Molecular Devices, Sunnyvale, CA, USA).

### 4.4. Western Blot Analysis

Cell and tissue proteins were extracted from cell lysates (Solarbio, Beijing, China) and the protein contents was determined with the BCA method (Beyotime, Shanghai, China). Cell lysates were dissolved in SDS sample buffer and heated at 95 °C for 5 min. Protein samples were separated via reducing 8% (*w*/*v*) polyacrylamide gel electrophoresis and transferred to PVDF membranes. Using the corresponding antibodies, immunoreactive bands were visualized with a hypersensitive chemiluminescent reagent. Antibodies were purchased from Cell Signaling Technology Company (CST, Boston, USA), including anti-insulin receptor β (#3025), anti-phospho-insulin receptor β (Tyr1150/1151) (#3024), anti-AKT (#9272), anti-phospho-AKT (Ser473) (#9271), anti-Glut4 (#2213), anti-AS160 (#2670), and anti-phosphoAS160 (Thr642) (#8881). Protein-band densities were normalized to the β-tubulin content.

### 4.5. Real-Time Reverse Transcriptase-Polymerase Chain Reaction (RT-PCR)

Total mRNA from mouse tissues was extracted using TRIzol reagent (Transgen Biotech, Beijing, China). mRNA was reverse-transcribed into cDNA with gDNA Eraser according to the instructions of PrimeScript™ RT reagent kit (TaKaRa Biotechnology, Dalian, China). Real-time fluorescence quantification was performed using the Roche LightCycler^®^ 480 fluorescence quantitative system (Roche, Basel, Switzerland) and expression of target genes determined using TB Green^®^ Premix Ex Taq™ II (TaKaRa Biotechnology, Dalian, China). In muscle tissue, α-tubulin was used as the endogenous control while in liver tissue and subcutaneous fat, β-actin was used. Gene levels were analyzed using the ΔΔCt method. The ratio of mRNA expression of the target gene to endogenous control α-tubulin/β-actin was 2−ΔΔC(T), ΔC(T) = C(T)target gene-C(T) α-tubulin/β-actin. Primer sequences are shown in Table 1.

### 4.6. Glucose Uptake Assay

The glucose uptake cell-based assay kit (No. 600470; Cayman Chemical, Ann Arbor, MI, USA) using 2-NBDG as the measurement standard was employed to detect cellular glucose uptake. C2C12 cells were cultured in black 96-well plates and after complete differentiation, treated with insulin (100 nM) and serpentine (10 μM) in glucose-free culture medium for 30 min. 2-NBDG at a final concentration of 100 µg/mL was added 10 min before the end of the treatment period. Culture plates were subsequently centrifuged at 400× *g* at room temperature. The supernatant was aspirated, followed by the addition of 200 μL of cell-based assay buffer to each well. Next, culture plates were centrifuged at 400× *g* at room temperature, the supernatant aspirated, and 100 μL cell-based assay buffer per well finally added. 2-NBDG uptake by cells was determined with fluorescent filters designed to detect fluorescein (excitation/emission = 485/535 nm).

### 4.7. SPR Analysis

The Biacore S200 instrument (GE Healthcare, Boston, MA, USA) was used to explore the interactions between serpentine and insulin receptor. The extracellular domain of IR (10 µg/mL) was immobilized on a CM5 sensor chip (GE Healthcare) in 10 mM sodium acetate at pH4.5 and serpentine (0.78125–25 μM) was passed through. The flow rate of IR fixed on the sensor chip surface was 30 μL/min, and binding and dissociation times were 90 s. Kinetic and affinity analyses were conducted using Biacore S200 software.

### 4.8. Statistical Analysis

Data are expressed as mean ± standard error of mean (SEM) of independent experiments. Statistical differences were calculated with unpaired t-test or post-hoc one-tailed Mann–Whitney U tests using Prism 7 (GraphPad Software, San Diego, CA, USA). Probability (*p*) values < 0.05 were considered statistically significant.

## 5. Conclusions

In conclusion, serpentine acts as a sensitizer to enhance the effect of insulin through interactions with the insulin receptor and improves glucose metabolism in both C2C12 and 3T3-L1 cells and HFD/STZ-induced diabetic mice. Mechanistically, serpentine activates AMPK-AS160-GLUT4 signaling to enhance glucose metabolism in C2C12 cells (Figure 7). While the detailed underlying mechanisms remain to be clarified in vivo, the current findings highlight the potential of serpentine as an adjuvant treatment for T2DM.

## Figures and Tables

**Figure 1 pharmaceuticals-16-00016-f001:**
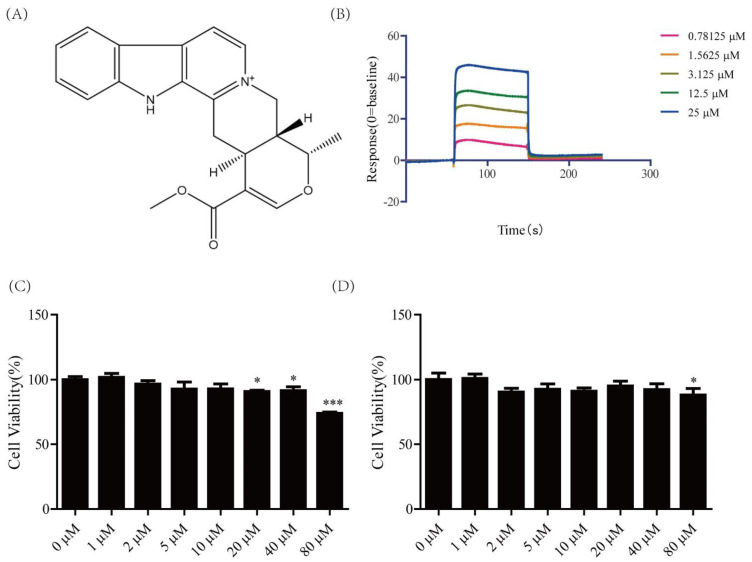
Cytotoxicity of serpentine and affinity of serpentine for the insulin receptor. (**A**) Chemical structure of serpentine. (**B**) Affinity of serpentine for the insulin receptor. (**C**) Cytotoxicity of ser-pentine against 3T3-L1. (**D**) Cytotoxicity of serpentine against C2C12. Data are expressed as means ± SEM (*n* = 6 in each group). * *p* < 0.05, *** *p* < 0.001 versus 0 μM group.

**Figure 2 pharmaceuticals-16-00016-f002:**
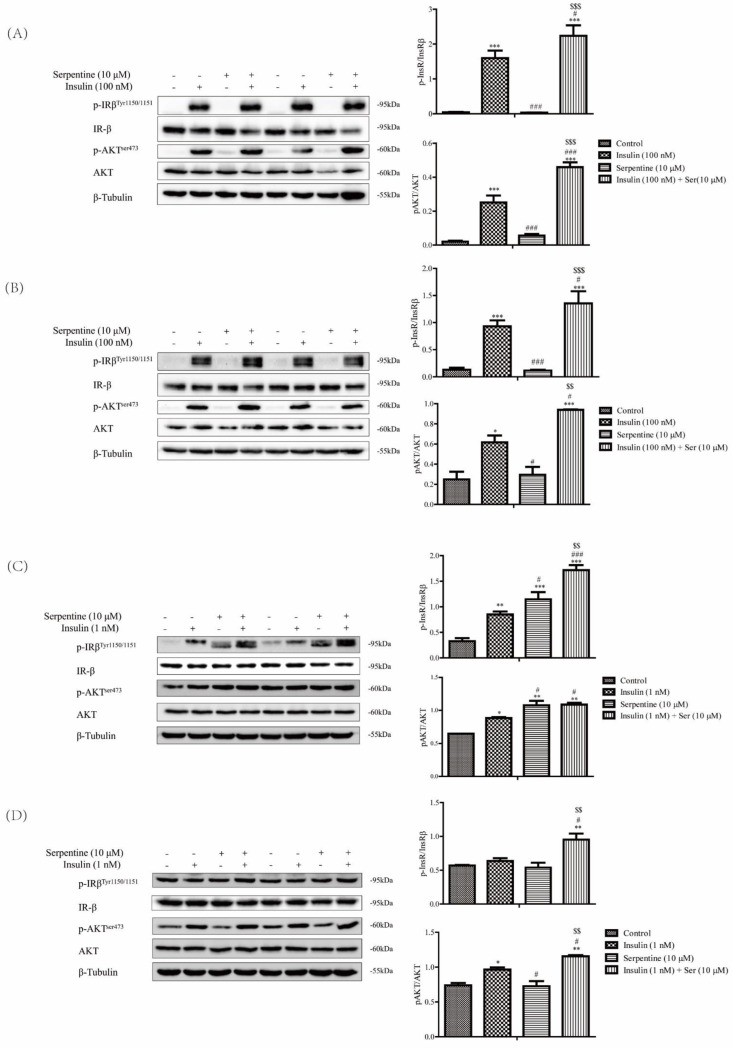
Effects of serpentine on the insulin signaling pathway activated by low-dose insulin. (**A**) Expression of key proteins of the insulin pathway in 3T3-L1 cells following treatment with insulin (100 nM) and serpentine (10 μM). (**B**) Expression of key proteins of the insulin pathway in C2C12 cells following treatment with insulin (100 nM) and serpentine (10 μM). (**C**) Expression of key proteins of the insulin pathway in 3T3-L1 cells following treatment with insulin (1 nM) and serpentine (10 μM). (**D**) Expression of key proteins of the insulin pathway in C2C12 cells following treatment with insulin (1 nM) and serpentine (10 μM). Data are expressed as mean ± SEM (*n* = 4 in each group). * *p* < 0.05, ** *p* < 0.01, *** *p* < 0.001 versus control; # *p* < 0.05, ### *p* < 0.001 versus insulin (1 nM) or insulin (100 nM); $$ *p* < 0.01, $$$ *p* < 0.001 versus serpentine (10 μM).

**Figure 3 pharmaceuticals-16-00016-f003:**
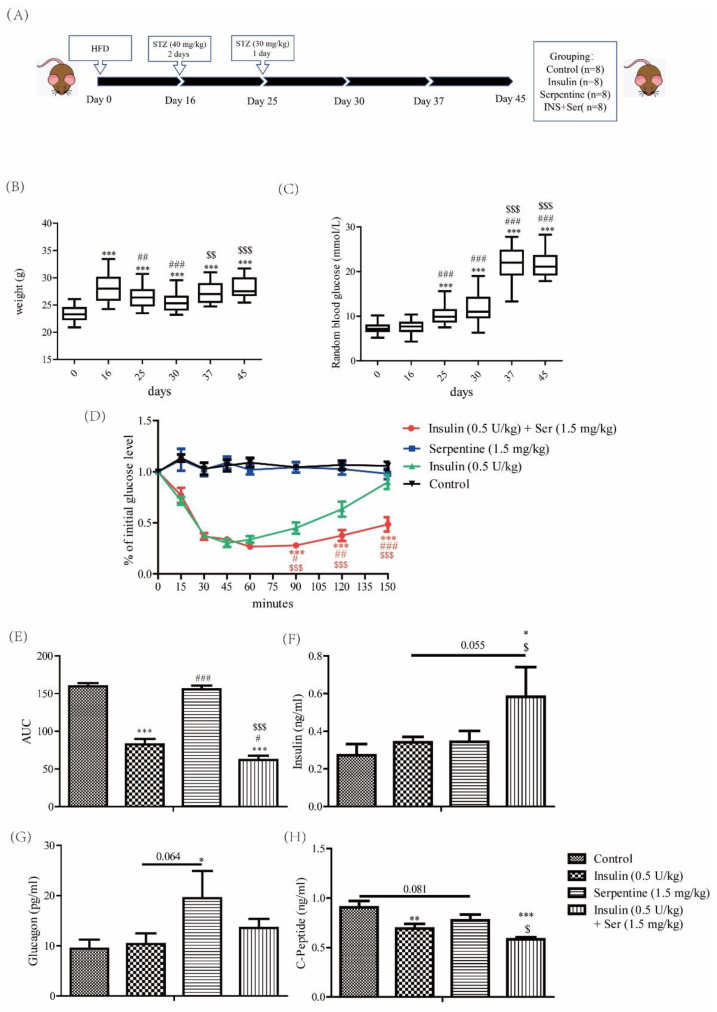
Effects of serpentine on blood glucose and hormone levels in an insulin-treated STZ-HFD-induced type 2 diabetes mouse model. (**A**) Time schedule for establishing the HFD/STZ-induced diabetic mouse model. Changes in body weight (**B**) and blood glucose (**C**) of mice during establishment of the animal model. Blood glucose changes within 150 min after insulin (0.5 U/kg) (**D**) and serpentine (1.5 mg/kg) (**E**) treatment, and AUC calculation. (**F**) Serum insulin, (**G**) glucagon, and (**H**) C-peptide levels in diabetic mice at 150 min. Data are expressed as mean ± SEM (*n* = 8 in each group). In (**B**,**C**), *** *p* < 0.001 versus 0 days; ## *p* < 0.001, ### *p* < 0.001 versus 16 days; $$ *p* < 0.01, $$$ *p* < 0.001 versus 30 days. In (**D**,**H**), * *p* < 0.05, ** *p* < 0.01, *** *p* < 0.001 versus control; # *p* < 0.05, ## *p* < 0.001, ### *p* < 0.001 versus insulin (0.5 U/kg); $ *p* < 0.05, $$ *p* < 0.01, $$$ *p* < 0.001 versus serpentine (1.5 mg/kg).

**Figure 4 pharmaceuticals-16-00016-f004:**
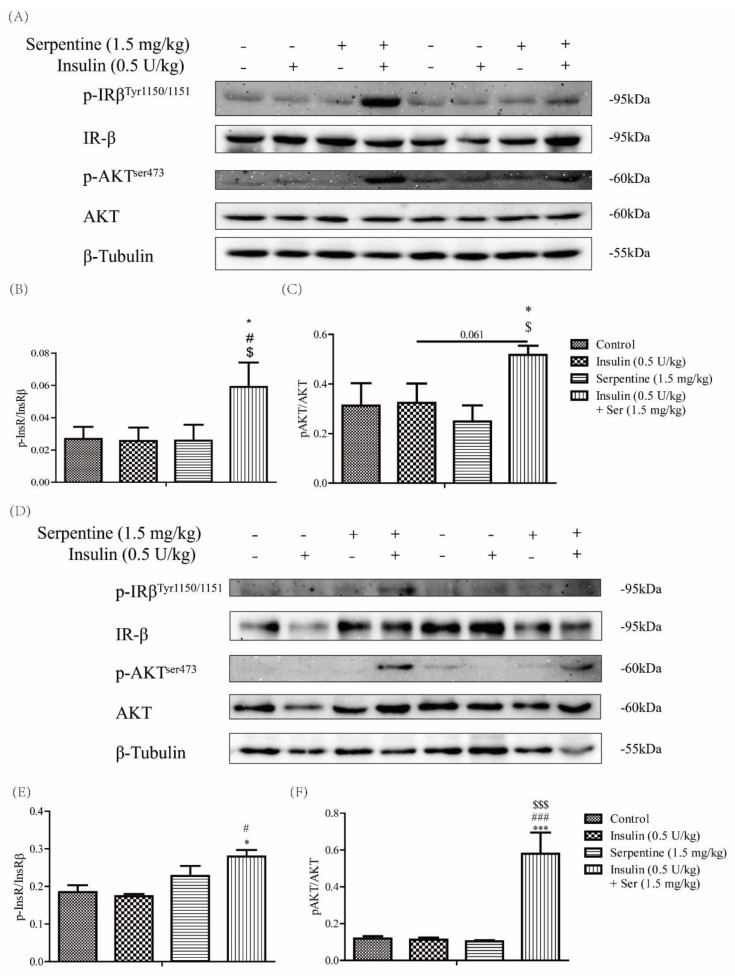
Effects of serpentine and insulin treatment on insulin signaling pathways in target organs. (**A**–**C**) Expression- and gray-level analysis of key proteins of the insulin pathway in muscle tissue. (**D**–**F**) Expression- and gray-level analysis of key proteins of the insulin pathway in subcutaneous adipose tissue. Data are expressed as mean ± SEM (*n* = 4 per group). * *p* < 0.05, *** *p* < 0.001 versus control; # *p* < 0.05, ### *p* < 0.001 versus insulin (0.5 U/kg); $ *p* < 0.05, $$$ *p* < 0.001 versus serpentine (1.5 mg/kg).

**Figure 5 pharmaceuticals-16-00016-f005:**
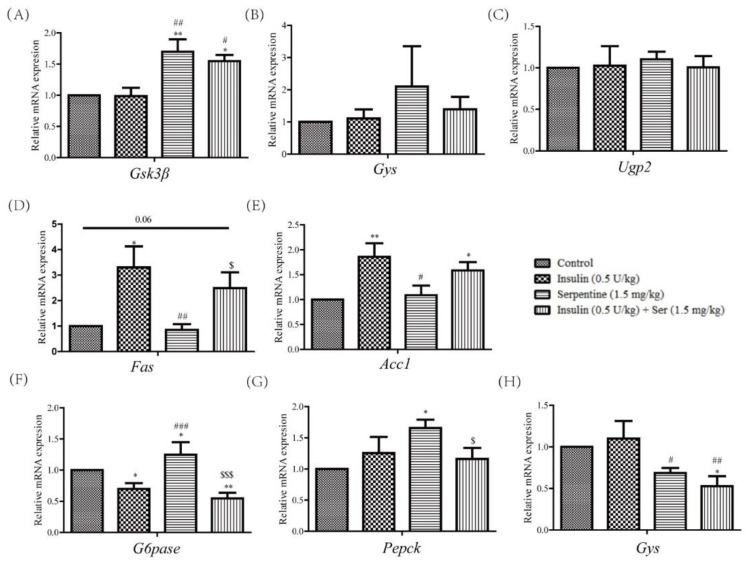
Effects of serpentine and insulin treatment on genes related to gluconeogenesis, glycogen synthesis, and fat synthesis. mRNA levels of glycogen synthesis-related genes (**A**) *Gsk3β*, (**B**) *Gys* and (**C**) *Ugp2* in muscle tissue. mRNA levels of fat synthesis-related genes (**D**) *Fas* and (**E**) *Acc1* in subcutaneous adipose tissue. mRNA levels of gluconeogenesis and glycogen synthesis-related genes (**F**) *G6Pase*, (**G**) *Pepck,* and (**H**) *Gys* in liver tissue. Data are expressed as mean ± SEM (*n* = 8 per group). * *p* < 0.05, ** *p* < 0.01 versus control; # *p* < 0.05, ## *p* < 0.01, ### *p* < 0.001 versus insulin (0.5 U/kg); $ *p* < 0.05, $$$ *p* < 0.001 versus serpentine (1.5 mg/kg).

**Figure 6 pharmaceuticals-16-00016-f006:**
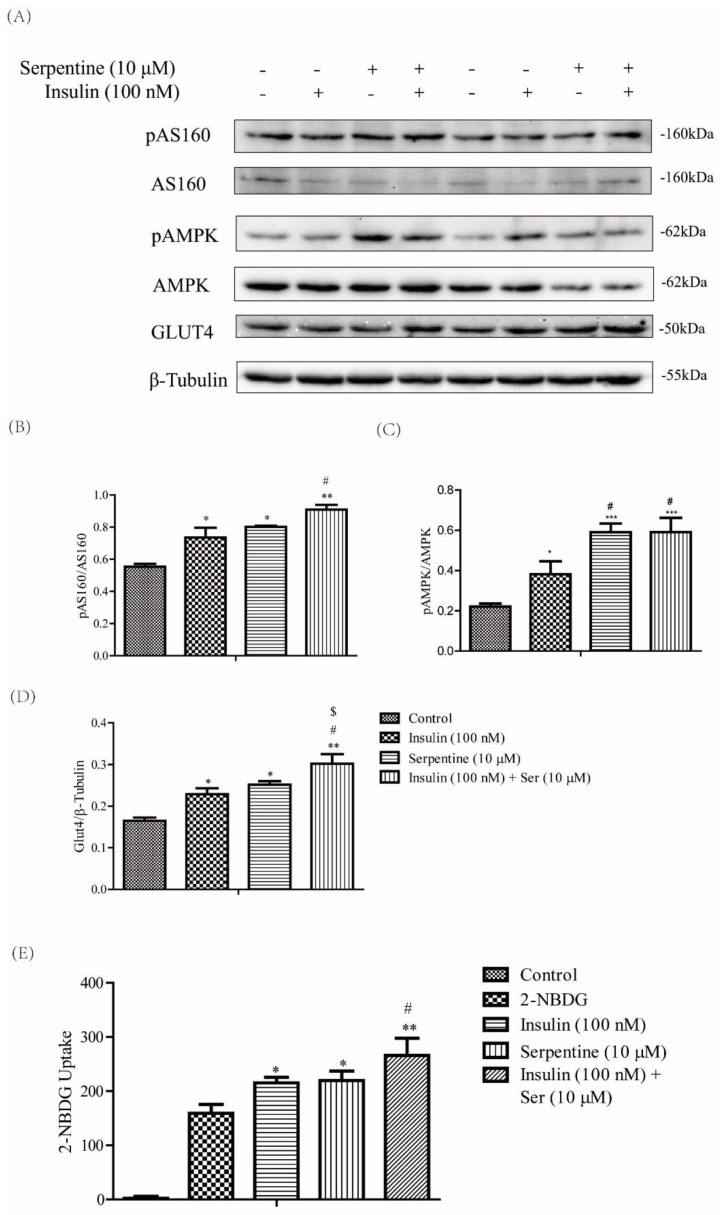
Serpentine promotes glucose uptake through AMPK signaling. (**A**) Expression of key pro-teins of the AMPK-induced glucose uptake pathway in C2C12 cells after treatment with insulin (100 nM) and serpentine (10 μM). (**B**) pAS160/AS160, (**C**) pAMPK/AMPK, (**D**) Glut4/β-tubulin, and (**E**) 2-NBDG uptake in C2C12 cells treated with insulin (100 nM) and serpentine (10 μM). Data are ex-pressed as mean ± SEM (*n* = 4 per group). * *p* < 0.05, ** *p* < 0.01, *** *p* < 0.001 versus control; # *p* < 0.05 versus insulin (100 nM); $ *p* < 0.05 versus serpentine (10 μM).

**Figure 7 pharmaceuticals-16-00016-f007:**
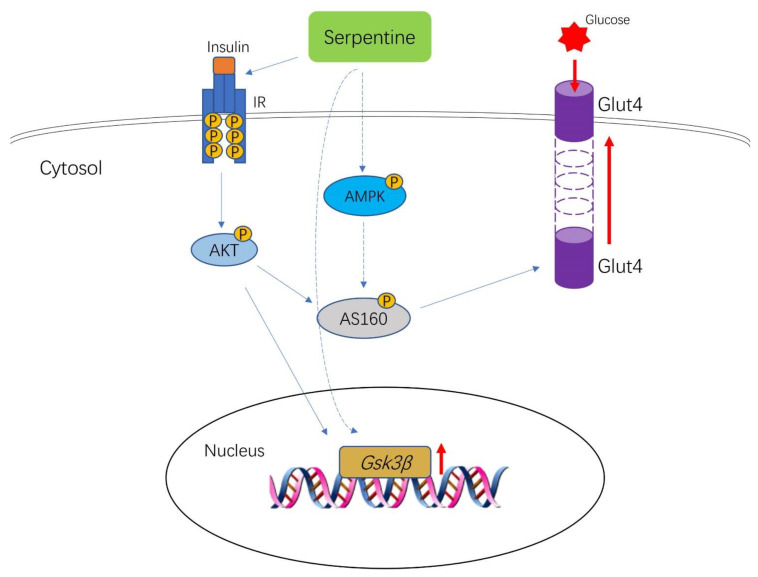
Serpentine acts as an insulin sensitizer to enhance its activation of the insulin signaling pathway through binding insulin receptor, improving glucose metabolism in C2C12 and 3T3-L1 cells and HFD/STZ-induced diabetic mice. Serpentine additionally activates AMPK-AS160-GLUT4 signaling, in turn, enhancing GLUT4 membrane transfer and improving glucose metabolism in C2C12 cells.

**Table 1 pharmaceuticals-16-00016-t001:** Gene sequences of primers.

Gene Name	Forward Primer	Reverse Primer
*β-Actin*	5′-*GAGACCTTCAACACCCCAGC*-3′	5′-*ATGTCACGCACGATTTCCC*-3′
*Gys*	5′-*ATCTTCTTCGTCTTCCGCATC*-3′	5′-*GACACTGAGCAGGGCTTTTCC*-3′
*G6pase*	5′-*AAAAAGCCAACGTATGGATTCCG*-3′	5′-*CAGCAAGGTAGATCCGGGA*-3′
*Pepck*	5′-*TTTGATGCCCAAGGCAACTT*-3′	5′-*ATCGATGCCTTCCCAGTAAA*-3′
*Fas*	5′-*CTGGCATTCGTGATGGAGTC*-3′	5′-*TGTTTCCCCTGAGCCATGTA*-3′
*Acc1*	5′-*CGCTCGTCAGGTTCTTATTG*-3′	5′-*TTTCTGCAGGTTCTCAATGC*-3′
*Ugp2*	5′-*TAACCAAGGGCACTGTAGGGA*-3′	5′-*GGAGCTGCAATTAAAAGTTTCG*-3′
*Gsk3β*	5′-*TCCATTCCTTTGGAATCTGC*-3′	5′-*CAATTCAGCCAACACACAGC*-3′
*α-Tubulin*	5′-*CCACAAGTTTGATCTGTTGCAT*-3′	5′-*GCAGCAACTAGTATCCCTGTCC*-3′

## Data Availability

The original contributions presented in the study are included in the article. Further inquiries can be directed to the corresponding authors.

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
