# Peer review of "Serpentine Enhances Insulin Regulation of Blood Glucose through Insulin Receptor Signaling Pathway"

_pharmaceuticals, 2022, doi:10.3390/ph16010016_

Round 1

Reviewer 1 Report

Peer review report on “Serpentine Enhances Insulin Regulation of Blood Glucose Through Insulin Receptor Signalling Pathway”

Manuscript ID: pharmaceuticals-2058753

This paper describes the anti-diabetic activity investigations carried out on serpentine, an alkaloid derived from Cantharanthus roseus (L.) D. Don. In general, the paper is well-written, well-described and the scientific investigations are thorough and competently performed. However there are some issues that need to be addressed.

Some comments:

Line 15.            You imply that periwinkle is a different plant to Cantharanthus roseus (L.) G. Don, but it is just the common name for the same plant. Please rewrite.

Line 20.            Capitalize the first letter of “western”.

Line 29.            Capitalize the first letter of “serpentine”.

Line 31.            Expand HFD/STZ the first time it is used.

Line 37.            Rewrite the sentence starting with “therefore” to indicate that serpentine may be useful as an insulin sensitizer.

Line 53.            What does AKT mean? Please expand.

Line 62.            Replace “drug” with plant.

Line 66.            Vincristine and vincristine presumably should be vincristine and vinblastine.

Lines 67-69.      Reference 19 talks about the ability of endophytes to influence the production of serpentine in the roots of Cantharanthus roseus which is commonly known as periwinkle. They are the same plant. Please rewrite these two sentences.

Line 68.            Reference 20 is nothing about serpentine being used in the treatment of hypertension and heart disease. It is simply an analytical paper about the isolation of alkaloids. Please address this.

Line 69.            You mention a previous study done by your group about serpentine. Please reference this report.

The citations for references 21 and 22 are missing from the text although I note that reference 21 is cited on line 271. This requires attention.

Line 128.          (B) in Figure 2 should be insulin (100 nM).

Line 320.          In materials and methods, please report how you acquired the serpentine. Was it sourced from a chemical company or isolated from the raw plant material?

Author Response

Dear reviewer, thank you very much for your valuable comments. I have revised the article one by one according to the questions you raised. Please see the attachment for specific answers.

Reviewer 2 Report

In this manuscript, authors reported that serpentine enhances insulin regulation of blood glucose through insulin receptor signaling pathway. All experiments were carefully performed and conducted to draw the conclusion. However, there are lots of problems and lacks as following:

1.      Authors must include line numbers for the convenience of review.

2.      Catharanthus roseus belongs to the class of periwinkle. Please correct the second sentence of the abstract.

3.      In the 3rd and 16th lines of the abstract, authors have to use capital letter for the first letter of serpentine.

4.      Authors should write the original name of the abbreviation “HFD/STZ” in the 18th line of the abstract.

5.      For the abstract, authors mentioned too much about the experiment contents and methods. Please rewrite mainly about the results and conclusions of the experiments.

6.      In the keywords, please remove insulin because insulin is a positive control.

7.      In the lines 20-21 of page 2, the source of two alkaloids should be recorded as “vinblastine and vincristine isolated from C. roseus”.

8.      In the line 21 of page 2, please correct typo for “vincristine and vincristine”

9.      In the line 24 of page 2, Catharanthus roseus (L.) G. Don à C. roseus

10.  In the line 24 of page 2, surface plasmon resonance à SPR or surface plasmon resonance (SPR)

11.  Authors should change the subtitle 2.1. as “Cytotoxicity of serpentine and affinity of serpentine for insulin receptor”.

12.  For the legend of figure 1, please make correction “Cytotoxicity and affinity of serpentine for the insulin receptor. (A) Structural formula of serpentine” to “Cytotoxicity of serpentine and affinity of serpentine for the insulin receptor. (A) Chemical structure of serpentine”.

13.  Why do one band and two bands coexist in p-IRβ of the figure 2 (C).

14.  There is a lot of difference between the first and second data of the negative control group (- / -) in the figure 2 (C), and in this case, it can cause many problems in the reliability of other data. Therefore, a reexamination is deemed necessary.

15.  In the line 1 of page 5, please unify STZ-HFD to HFD/STZ.

16.  For the figure 3 (A), authors should redraw the administration schedule.

17.  In the (+ / +) group of p-IRβ in figure 4 (A), there is a big difference between the first and second data, so reexamination is necessary.

18.  The data of GLUT4 in figure 6 (A) cannot be confirmed due to the contrast of the picture. Authors should adjust the brightness of the picture or conduct a re-experiment.

19.  In the figure 6 (A), data of pAS160 and AS160 also not clear.

20.  Please clarify the expression for the administration groups in the figure 6. (+ 2-NBDG)

21.  Please correct the typo-error “that that” in the line 12 of page 8.

22.  Please note the administration on “treated C2C12 cells” in the line 11 of page 8.

23.  Authors should add a brief description for TLK16998 in the line 12 of page 9.

24.  Authors should write the source of insulin and serpentine in the “materials and methods”.

25.  In the lines 13th and 15th of page 11, authors wrote 40mg/kg and 30mg/kg STZ as dose. However, Figure 3 (A) showed 35mg/kg STZ as dose. Thus, authors have to make correction about that.

26.  In the line 15th of page 14, please describe the frequency and duration for administration of STZ.

27.  The frequency and duration of administration for insulin and serpentine should be described in the “materials and methods”.

28.  It is necessary to explain the differences and actions of serpentine and insulin receptors related to diabetes. In addition, it is thought that it is necessary to provide evidence and explanations as to whether the effects of serpentine presented in this manuscript appears only as an action on the insulin receptor.  

Author Response

(The authors gave the same response as above.)
